# Lumbar Sitting Behavior of Individuals with Low Back Pain: A Preliminary Study Using Extended Real-World Data

**DOI:** 10.3390/s24206751

**Published:** 2024-10-21

**Authors:** Frederick A. McClintock, Andrew J. Callaway, Carol J. Clark, Raee S. Alqhtani, Jonathan M. Williams

**Affiliations:** 1Department of Rehabilitation and Sport Sciences, Bournemouth University, Bournemouth BH1 1PH, UK; fmcclintock@bournemouth.ac.uk (F.A.M.); acallaway@bournemouth.ac.uk (A.J.C.); cclark@bournemouth.ac.uk (C.J.C.); 2School of Applied Medical Science, Najran University, Najran P.O. Box 1988, Saudi Arabia; rsalhyani@nu.edu.sa

**Keywords:** accelerometer, spine, posture, variability, pain, acceptability

## Abstract

Low back pain affects 619 million people worldwide and is commonly provoked by sitting. Current assessment methods constrain task variability, removing real-world, task-switching behaviors. This study utilized accelerometers to provide an original validated method of in vivo real-world assessment of lumbar sitting behavior throughout a full day. A three-stage study design was used, which involved (1) blinded verification of our sitting detection algorithm, (2) full-day data collection from participants with low back discomfort, quantifying lumbar angles, and end-user acceptability explored, (3) case study application to two clinical low back pain (LBP) patients, incorporating measurement of provocative sitting. Focus group discussions demonstrated that data collection methods were acceptable. Sitting ‘windows’ were created and analyzed using novel histograms, amplitude probability distribution functions, and variability, demonstrating that sitting behavior was unique and varied across individuals. One LBP patient demonstrated two frequent lumbar postures (<15% flexion and ~75% flexion), with pain provocation at 62% lumbar flexion. The second patient demonstrated a single dominant posture (~90% flexion), with pain provoked at 86% lumbar flexion. Our in vivo approach offers an acceptable method to gain new insights into provocative sitting behavior in individuals with LBP, allowing individualized unconstrained data for full-day postures and pain provocation behaviors to be quantified, which are otherwise unattainable.

## 1. Introduction

Low back pain (LBP) remains the greatest cause of years lived with disability and directly affects 619 million people worldwide [1]. Prevalence estimates are projected to increase, demonstrating LBP remains a significant and growing societal challenge [1]. Epidemiological studies have demonstrated a link between sedentary sitting time and LBP [2,3,4]. These links are associational, leaving the question of causation unclear [5]. However, sitting is reported as one of the most common aggravating factors for people with low back pain [6,7,8].

Previous studies have compared sitting posture between individuals with LBP and those without, with some demonstrating differences in lumbar angle [9,10] and some showing no postural differences [11,12]. One of the challenges of studying sitting, and its association with LBP, is the application of laboratory-based studies to the real world. Previous studies of lumbar posture [13,14,15] have created highly stylized situations with up to 2 h of laboratory-based sitting to provoke pain. Such a constraint of task can be described using the established framework of variability from Cowin et al. [16], where outcome variability is completely removed (participants complete an identical singular task in sitting), which then reduces the execution variability (the variability associated with completing the task). For all participants, this reduces an individual’s ability to replicate daily living tasks where there would be typical switching between tasks, which would naturally allow breaks in sustained sitting. Task switching would allow the individual opportunities for different postures, thus, potentially creating false conclusions for the mechanism of action to provoke pain in laboratory-based tasks.

In addition to sitting posture, a recent systematic review found that sitting time (or duration) was associated with pain, providing evidence that sitting duration is linked to LBP [17]. Therefore, the limited duration of postural assessment, common in the literature and clinical practice (typically from seconds to minutes), provides minimal insights into the relationship between posture and duration of sitting throughout daily living. To understand the time-varying nature of lumbar posture, new methods of real-world, extended data collection are required.

To study the interaction between posture and duration, in the real world, rigorous methods for prolonged data collection are necessary. Camera-based systems, which are typically used, are limited to a specific data collection space. Technological development has led to an increasing trend of more portable devices such as electromagnetic trackers or inertial measurement units (IMUs) allowing new data collection possibilities [18]. However, IMUs are affected by the presence of metals, which are likely to be highly prevalent in office environments [19,20,21]. IMUs running a fusion of gyroscopes and accelerometers are a potential solution but the presence of gyroscopic drift over prolonged periods is likely to be a challenge for full-day data collection. Our laboratory has validated accelerometers for sagittal plane angle analysis [22], and with the benefit of being small, non-intrusive, cheap, and robust to the data collection environment, this approach could serve as the ideal solution for long-term measurement of posture and sitting duration.

Prior to such methods being recommended, it is crucial to verify the algorithms within daily living and determine the acceptance of the methods within the population, as well as explore the novel analysis methods now possible due to the extended real-world data collection. This would result in a step change in understanding the interaction between lumbar posture and its temporal components, providing original insights into sitting behavior in the real world, with the removal of task constraints, within daily free living.

Therefore, the aim of this study is to explore real-world sitting behavior in individuals with LBP and how this relates to the provocation of pain during sitting to significantly enhance knowledge in the field. Several interdependent studies are required to achieve this aim, namely:Study 1: Verification of the sitting detection algorithm to characterize the lumbar sitting behavior, with a blinded assessor.Study 2: Day-long assessment of the sitting behaviors of individuals with sitting related low back discomfort and feature extraction with an exploration of participant experiences.Study 3: Case study application to individuals with clinical LBP.

Prior to applying a whole-day measurement method to patients, it is imperative that sitting can be automatically identified and extracted from the sensor data. Once achieved, offering original ways of visualizing and summarizing data is necessary. It is also imperative to explore the experiences of people wearing the sensors for the whole day to minimize barriers to acceptance, verify the sitting detection algorithms, and establish a novel presentation of the sitting behavior data. If acceptability is low within the desired population, then the method may not be appropriate to apply to patients. Therefore, the primary objective was to verify the sitting detection algorithm and explore original ways of analyzing and summarizing the prolonged sitting behavior data. The secondary objective was to explore the acceptability of wearing the sensors throughout a whole day.

All studies were conducted according to the guidelines of the Declaration of Helsinki and approved by the Ethics Committee of Bournemouth University. Informed consent was obtained from all participants involved in the study.

Study 1: Verification of the Sitting Detection Algorithm using a Blinded Assessor.

## 2. Materials and Methods

The sensor setup utilized in this study has been previously validated [22] for angle calculation; however, this verification study was needed to determine the functionality of automatically detecting sitting from a whole day of kinematic data. Correctly detecting the sitting periods within real-world daily-living conditions is essential to be able to characterize lumbar sitting behavior.

### 2.1. Design and Participants

A cross-sectional, observational study design was used, utilizing a single participant (male, 80 kg, 1.73 m) with a blinded outcome assessor. Participant inclusion criteria included self-declared good health, not seeking any treatment for leg or back pain, and no known musculoskeletal disorders of the back or lower limb.

### 2.2. Sensor Placement and Data Processing

Three inertial measurement units (Movella Xsens Dots, Enschede, The Netherlands) were attached to the skin over the L1 and S2 spinous processes, and the lateral aspect of the right thigh, midway between the lateral epicondyle and greater trochanter, over the iliotibial band (Figure 1A). To ensure the attachment of the S2 sensor and minimize the effect of clothing, a 3D-printed mount was developed (Figure 1B).

Accelerometer data were collected at 15 Hz and imported into a custom Matlab script (Matlab 2023a, Mathworks, Natick, MA, USA) that derived the absolute and relative angle (between two adjacent sensors) using the ATAN2 function, having corrected the initial orientation of each sensor [22]. The angles were calculated for the sagittal plane only, representing the spine and hip flexion and extension. All data were filtered using a 4th order low-pass Butterworth filter with a 1 Hz cutoff [23]. Sagittal angle data were individually normalized to peak flexion range of motion (ROM) and expressed as a percentage of flexion ROM.

Once sensors were attached, the participant completed 3 lumbar-hip flexion-extension trials in standing, and then a bespoke set of sitting and standing tasks, each for around 2 min (Table 1). The tasks were designed to replicate different sitting postures adopted by individuals e.g., cross-legged sitting, interspersed with standing, walking, or lying tasks. The assessor running the Matlab script (outcome assessor) was blind to the tasks, the duration of tasks, and the order of tasks until data processing was complete. They were also absent during the period of data capture.

**Table 1 sensors-24-06751-t001:** The order and times of tasks completed in the blinded verification study.

Time	Activity	Period of Sitting (Figure 2)
0–2 min	Standing	
2–4 min	Flexion and extension cycles × 3 then standing	
4–6 min	Sitting	1
6–8 min	Standing	
8–10 min	Slouched sitting	2
10–12 min	Walking	
12–14 min	Cross-legged sitting slouched—both legs on chair	3
14–16 min	Standing	
16–18 min	Cross-legged sitting erect—both legs on chair	4
18–20 min	Walking	
20–22 min	Flex and extension cycles × 3, then standing with pause	
22–24 min	Sitting cross-legged, right over left	5
24–26 min	Standing	
26–28 min	Sitting cross-legged, left over right	6
28–30 min	Prone lying	
30–32 min	Supine lying	
32–34 min	Supine lying with legs raised	
34–36 min	Standing	
36–38 min	Sustained forward bending	
38–40 min	Highly variable sitting	7
40–42 min	Sit to stand	
42–44 min	Standing on one leg in the Trendelenburg posture, right	
44–46 min	Full kneeling with head to floor, hold for 15 s, repeat × 3	
46–48 min	High stool sitting	
48–50 min	Erect sitting	8

min: minutes; s: seconds.

**Figure 2 sensors-24-06751-f002:**
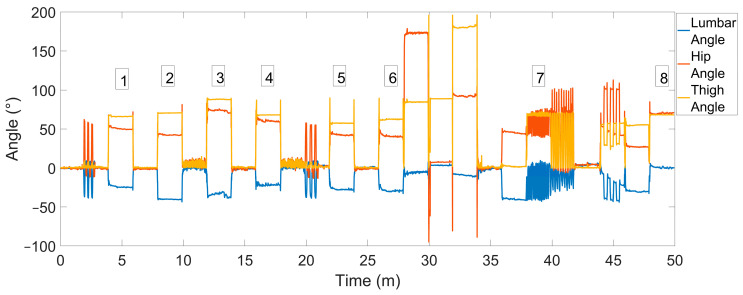
The time trace of the lumbar spine, hip and thigh angle using the three-sensor setup. Numbered to show periods of sitting.

## 3. Results

The sitting detection algorithm correctly identified seven of eight sitting periods throughout the data, where it missed region 5 (Table 1 and Table 2, and Figure 2). The results demonstrate that regardless of the lumbar sitting posture (slouched or erect, regions 2 and 8), leg posture (single crossed or double crossed, regions 3, 4, and 6), or degree of fidgeting/shifting (still or variable posture, regions 1, 2, 7, and 8), sitting was correctly identified from the data. The only section of sitting missed by the detection algorithm was region 5 (sitting cross-legged, right over left) where the thigh sensor was 3° below the window of detection. This suggests that the cross-legged position of the ankle to the opposite knee may not trigger sufficient hip flexion to signify sitting. The algorithm was amended to reduce the threshold. The script was re-run to ensure this sitting window was correctly identified and no impact on the identification of other sitting periods was observed.

Study 2: Day-long assessment of sitting behaviors in individuals with sitting related low back discomfort.

## 4. Materials and Methods

### 4.1. Design and Participants

A cross-sectional, observational study design was used. Six participants volunteered for this part of the study (age 29 ± 5.2 years, height 178.6 cm ± 9.6 cm, weight 78.5 kg ± 17.4, sex M = 3 and F = 3). Participant inclusion criteria included self-declared good health, not seeking any treatment for leg or back pain, and no known musculoskeletal disorders of the back or lower limb. It was important to explore the acceptability and data in individuals similar to the target population; therefore, people with some low levels of sitting discomfort were utilized.

### 4.2. Sensor Placement and Data Processing

Participants wore three IMUs (Figure 1) throughout a whole day, following the same attachment procedure as Study 1. Participants attended a mutually agreed space, where the sensors were attached by a single researcher, and recording was started. Participants were guided through three cycles of full lumbar flexion and extension in standing, and fully slouched to fully erect sitting, which allowed the full range of motion to be determined. They were then encouraged to carry out their normal daily activities, while wearing the sensors. The only constraint was to avoid submersion in water during the data collection period. At the end of the day, the participants returned to the laboratory, the sensors were removed, and the data were downloaded.

### 4.3. Data Analysis

#### 4.3.1. Focus Group

After all testing had been completed, the six participants were invited to take part in a participant involvement focus group, to determine the experience of wearing the sensors for the day. The participant involvement focus group was thematically analyzed by two assessors to generate key findings from the discussions. These were generated into a key points category and a future recommendations category.

#### 4.3.2. Visualization and Feature Extraction

Each of the six participant’s whole-day data (eight hours) were transferred to Matlab where custom algorithms produced angles and windows of when the participants were sitting [22]. The sitting detection algorithm uses inputs from the absolute angle of the thigh sensor, in addition to the relative angles between the thigh and S2 sensor (hip), and L1 and S2 sensor (lumbar spine) to determine when sitting was occurring (Study 1). Sitting windows were defined as sections of sitting data greater than 1 min, where multiple kinematic conditions and time were met. This allowed for the analysis of the sitting behavior within each sitting window, where the duration, mean, and standard deviation of the lumbar angle were calculated.

The mean represents the average lumbar angle used during that window, and the standard deviation represents the degree of ‘fidgeting’ or variability around that mean lumbar angle [24].

To provide visualization of the multiple sitting windows across the day, a novel histogram was created, where the bar height represents the mean lumbar angle (as a percentage of the total flexion range), the error bar represents the standard deviation of lumbar posture during that sitting period, and the width of the bar represents the duration for that sitting period.

As there is a likely relationship between sitting posture and duration, an original variable, sitting lumbar posture exposure (*SLPE*), was used to quantify the interaction between the lumbar angle, duration, and fidgeting, using Equation (1),
(1)$$SLPE=\frac{\left(T*F\%\right)}{SD\_pos}$$
where *T* is time in minutes for that sitting posture window, *F*% is the sagittal angle of the lumbar spine expressed as a percentage of the total flexion range of motion (established at the start of the day’s data collection), and *SD_pos* is the standard deviation of the lumbar sagittal angle for that sitting window.

Sitting lumbar posture exposure (*SLPE*), shows a high value when an individual spends a long duration at a high percentage of their maximum lumbar flexion (or extension) range of motion, while also remaining stationary around this lumbar posture.

In addition to the per-sitting-window analysis, a full-day summary of the sitting data was calculated to provide a description of an individual’s ‘sitting signature’ for the day. The variables used to describe this include the following:

Average posture for the day, calculated as the unweighted mean of all the sitting-window means. This represents an individual’s average posture for the day from each postural window. If an individual had 10 sitting windows, then this would be the mean of these 10 mean postures.

Weighted average posture for the day, calculated as the weighted mean of all the sitting-window means. This represents an individual’s average posture for the day, considering the duration of each posture.

Variability of postures through the day, which is the standard deviation of the mean postures from each sitting window. This represents the variability of the postures used.

Average ‘fidgeting’ across the day, which is the mean of the standard deviations of each window. This represents the average ‘fidgeting’ or variability within the sitting window.

Average sitting lumbar posture exposure for the day, which is the unweighted average of *SLPE*.

Variability of *SLPE* through the day, which is the standard deviation of the *SLPE* means from each sitting window.

Weighted average *SLPE*, which is the weighted average version of *SLPE*, weighted by the duration of each sitting window.

The sitting behavior across time was also quantified using the methods described by Dunk and Callaghan [25]. This involves quantifying the lumbar sitting posture into 2% incremental ‘bins’ for every time point, resulting in an amplitude probability distribution function (APDF), where the *x*-axis represents the lumbar angle and the *y*-axis the frequency as a percentage of the total counts for the day. Such a method helps visualize the frequency of use of specific lumbar postures.

## 5. Results

All six participants completed the full day of data collection wearing the sensors through their normal workday (approx. 480 min). The mean overall time spent sitting throughout the day was 132.8 ± 61.0 min.

### 5.1. Focus Group

All six participants completed and contributed to the focus group discussion. The results of this discussion highlighted several key points, primarily focused on attachment anxiety and clothing. Participants were anxious that the sensors may fall off, particularly if knocked, and recommended a review of the attachment to consider some reinforcement. Only one sensor fell off during this study and it was the thigh sensor. In line with this, participants were concerned that clothing may knock the sensor and recommended advising individuals to wear loose-fitting clothing. Despite this, they reported that most of the time they forgot they were wearing the sensors and found wearing the sensor comfortable and unobtrusive. Greater time variance sessions were also suggested to include weekend days and more data collection during the evening to fully encapsulate the lumbar movement outside of typical working and daytime hours.

### 5.2. Visualization and Feature Engineering

APDF plots for each participant can be seen in Figure 3, along with the window-by-window analysis for the day for each participant in Figure 4. These results (Figure 3 and Figure 4) visually demonstrate that postural frequency is very individual. Similar shapes in the APDF plots (Figure 3) can be seen for P4 and P5, but not for the remaining four participants. P4 and P5 seem to demonstrate a very consistent ‘single posture’ shape with a posture between 65 and 80% (P4) and 80 and 95% (P5). This is confirmed by the average postures (Table 3, 70.91% and 81.34%, respectively) and low variability of posture (5.86% and 7.97%, respectively). A variability of 8% across the day suggests a very consistent postural signature.

The window-by-window analysis for P4 and P5 (Figure 4) demonstrates a relatively consistent height of bars representing the average lumbar angle; however, the sitting duration (bar width) of the windows demonstrates that P4 uses a lot of short sitting periods, whereas P5 has a single dominant period (green bar in Figure 4) around a series of short periods. In contrast, P1 and P2 utilized a wide variety of postures as demonstrated by the variability of posture of 18% and 25%, respectively, and greater variability in the shape of the APDF (Figure 3). The window-by-window plots (Figure 4) are chronological across time, therefore, P1 demonstrates more upright postures earlier in the day (left side) compared to later in the day (right side).

Study 3: Case studies of care-seeking individuals (application and classification).

## 6. Materials and Methods

The aim of this study was to provide an example application of the defined methods to two individuals seeking care for their LBP. An n = 2 case study design was used in a clinical population.

### 6.1. Participants

Two participants with physician diagnosed LBP were recruited for this final part of the study. Participants were screened by a physiotherapist to ensure inclusion criteria were met. Inclusion criteria were as follows: physician diagnosed LBP; pain confined to the region between the 12th rib and inferior gluteal folds; symptoms for at least three months, and pain evoked by postures or tasks throughout the day (movement evoked back pain). Individuals were excluded if they had symptoms below the gluteal folds including neurological symptoms. Average and worst pain in the preceding 2 weeks were collected using a visual analogue scale (VAS), fear of pain, and subsequent avoidance of movement was assessed using the fear-avoidance beliefs questionnaire (FABQ) [26], and disability was assessed using the Roland–Morris disability questionnaire (RMDQ) [27]. As proof of application, no formal sample size calculation was completed. Participants provided written informed consent prior to commencing the study.

The methodology was the same as that described above in Study 2. The only additions were that participants were informed to wear loose-fitting clothing as per the previous participant involvement focus group, and they were asked to note down any periods of discomfort or pain experienced in the low back region and a description of what they were carrying out at the time, as a diary.

### 6.2. Data Analysis

Whole-day data (eight hours) were transferred to Matlab where custom algorithms produced angles and sitting windows as outlined above [22]. Data were processed producing sitting windows for comparisons and whole-day summary outcomes.

The discomfort diary was digitalized and used to identify when an individual experienced a provocation of symptoms. These windows were deemed ‘painful windows’, and the data were separated to enable descriptive comparison between ‘painful windows’ and ‘non-painful windows’.

## 7. Results

The two participants (Table 4) completed the full day of data collection wearing the sensors through their normal workday. The average time sat through the day was 161.4 ± 31.2 min.

The sitting outcomes for each participant are displayed in Table 5 and Figure 5 and Figure 6. The APDF (Figure 5), which visualizes the frequency of different postures used, demonstrates that Pt1 has two distinct postures, one upright <15% lumbar flexion, and the other between 60 and 90% of flexion. In contrast, Pt2 has a distinct single posture around 90% of flexion and above, with very little posture variability. This is mirrored in the through-day window-by-window analysis (Figure 6) where, despite some upright postures early in the day, Pt2 utilizes these large flexion postures throughout the day, with small amounts of fidgeting. However, Pt1 uses a variety of postures throughout the day, with a few very upright sitting postures in the middle of the day.

Sitting postures that provoked pain are denoted in red (Figure 6). Postures where pain was evoked for Pt2 produced similar metrics to their typical posture, suggesting that pain provocation was associated with their usual habitual posture. Pt1 demonstrates a more varied pattern of pain provocation; however, with only three painful windows, each at quite different postures (30%, 50% and 75% full range of motion), which, based on their ADPF (Figure 5), suggests the 75% region is a highly frequent posture, with the other two being relatively infrequent postures. Both Pt1 and Pt2 (to a lesser extent) seem to be provoked by more flexed postures, with the difference being that Pt1 seems to have two habitual postures, where the more upright one (less flexed) might serve to minimize provocation. However, Pt2 seems to only have a single habitual posture, perhaps suggesting an inability to alter their posture from this single strategy.

## 8. Discussion

The aim of this study was to explore the real-world sitting behavior of individuals with LBP and how this relates to the provocation of pain during sitting. To achieve this, we successfully presented a series of interrelated studies demonstrating a novel method of obtaining real-world sitting behavior applied as a proof of concept to individuals with a clinical condition, LBP.

This study makes several original contributions to the literature. Firstly, it demonstrates that the method of employing accelerometers to measure lumbar and hip angles is a viable solution to obtaining whole-day lumbo-pelvic-hip kinematics and that sitting data specifically can be extracted. Previous studies have employed accelerometers for measuring sitting [25,28]; however, the sensors were limited to the lumbar spine only, which makes the identification of sitting difficult due to the lack of thigh/hip information. Thigh sensors have been employed successfully as activity monitors [29] but are less common in the monitoring of individuals with LBP. To this end, focus group discussions provided valuable information pertaining to the acceptability of such a configuration. Previous systematic reviews of wearables for individuals with LBP have not explored the concept of acceptability [30]. This is critical if the methods are to be integrated into routine clinical practice [31]. Therefore, the findings from this study suggest that participants were accepting of the wearables, and reporting that they forgot they were there for most of the time.

The second contribution relates to the opportunity provided by data across a prolonged time frame. Previous studies exploring sitting behavior in LBP are often limited to 2 h [14,15]. The methods proposed in this study would capture up to 8 h of sitting if the participant sat for the whole day. However, with such a wealth of data, original methods of summarizing and visualizing sitting behavior are necessary and are provided by this study. The window-by-window and APDF analysis offer clinicians a quick and meaningful summary and demonstrate how unique individuals are in relation to sitting. For example, as shown in Figure 3 and Figure 4, P1–P3 had varied sitting postures indicating more lumbar movement around sitting, with P4 and P5 having distinct sitting postures at a set percentage flexion indicating a more fixed and routine sitting posture. P6 had a specific flexion posture at a high flexion percentage, while also showing an increase in a more neutral lumbar position, suggesting an attempt to regularly sit more upright but commonly resulting in a more flexed posture. P2, P3 and P6 have varied sitting windows throughout the day with little pattern apart from P2 having large differences between neighboring windows, suggesting the notion of postural modification between windows. P1, throughout the day, has an ever-increasing lumbar flexion percentage, which suggests an increase in flexed postures. P4 and P5 show a consistent flexion percentage throughout the day with P5 exhibiting a more varied sitting-window length at this posture. These can also be compared to Figure 5 and Figure 6 of the LBP patients, where Pt1 demonstrates two frequent lumbar postures (<15% flexion and ~75% flexion), with pain provoked at an average of 62% lumbar flexion. The second patient demonstrates a single dominant posture (~90% flexion), with pain instances at an average of 86% lumbar flexion. This shows high variability within and between populations.

Previous studies, comparing sitting between individuals with LBP and those without, have focused specifically on a single snapshot in time, usually a laboratory assessment [32,33]. Findings describing individuals with LBP and demonstrating differences from those without LBP [9] have the potential to make erroneous conclusions. These snapshot assessments do not take into consideration postural variability. As seen in the data from this study, no ‘one posture’ represented any of the participants. Therefore, conclusions about individuals based on a single posture may be true, but only for that specific posture and for that specific individual. The sitting literature does contain studies of prolonged sitting, up to 2 h or so; however, these tasks are designed to constrain any task variability [14,25]. For example, a 2 h typing task represents one, very specific task, with a single outcome (typing). Such an approach provides strong constraints to execution variability, and, therefore, fails to capture the breadth of postures (variability) inherent in an individual’s sitting behavior. Conclusions based on such constraints may not adequately represent the range of sitting execution variability across the day.

The final contribution of this study is around variability. Variability relating to LBP is a less frequently studied construct compared to traditional group that summarize kinematics. Previous studies have shown differences in variability of kinematics, muscle activity, and sitting posture in individuals with LBP [34,35,36]. However, the current study showed that variability, as measured through the number of different postures used and the shape of the sitting windows and APDF profiles, was highly individualized. Some individuals had a dominant ‘posture’ spread across only 10–15% of the total flexion range, whereas others a much greater spread of posture (higher variability). As posture determines the level and distribution of tissue stress within the lumbar spine, reduced variability would result in the same underlying tissues being repeatedly stressed during sitting [37], as perhaps observed in Pt2. Conversely, high sitting variability would ‘share’ the load around different structures. Moreover, the fidgeting around the posture may also move the load around different structures, albeit in a temporary fashion. Pt1 had a tendency towards this, with large error bars within the window-to-window analysis, indicating a greater fidgeting behavior. Greene et al. [38] demonstrated that individuals who fidgeted less went on to develop LBP in a prolonged sitting task, suggesting fidgeting may offer some mediation of pain development. The current work demonstrates that fidgeting is highly individual, and somewhat variable likely to be due to non-constrained normal sitting.

The current work demonstrates a robust, validated, verified, and user-accepted, data collection method using accelerometers, with the removal of laboratory-based task constraints, allowing spinal postures to be quantified within real-world daily living. The original and significant development described within this work is the novel feature of extraction and application of visualization of through-day sitting postures. This application allows the development of posture variability within a clinical setting, developing the notion of postural variability within long-term data analysis, description of sitting posture across time in ways that quantify posture, and challenges existing notions of a single posture, which is often the result within laboratory-based assessment. A limitation of this proposed method is drawing information from the sagittal plane only. Therefore, information related to other planes is not considered, which could be relevant to other clinical conditions and should be investigated in future studies.

## 9. Conclusions

This original assessment and analysis method offers a real opportunity for future work to investigate new insights into the provocative sitting behavior of individuals with LBP, while also not being limited to LBP. This could be applied to different conditions to identify sitting characteristics in pain-free individuals or those with other clinical conditions such as, but not limited to, Stroke, Parkinson’s, and Fibromyalgia. The removal of task constraints allows data capture within real-world daily living. This opens the possibility for new approaches to LBP management through large-scale data collection analyzed with cluster analysis, to investigate novel methods to inform clinical decision making for LBP management strategies.

## Figures and Tables

**Figure 1 sensors-24-06751-f001:**
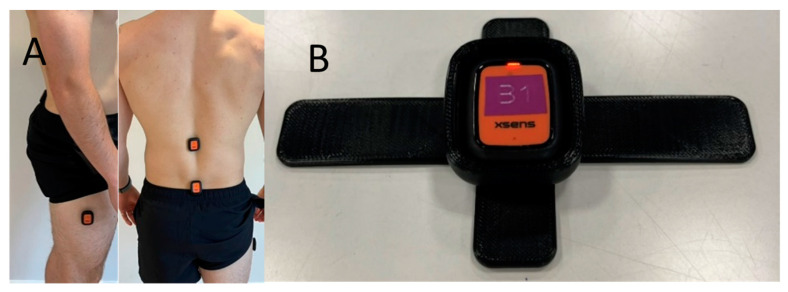
(**A**) The location of the three Xsens IMUs. (**B**) The Movella Xsens Dot in the 3D-printed mount for S2 attachment.

**Figure 3 sensors-24-06751-f003:**
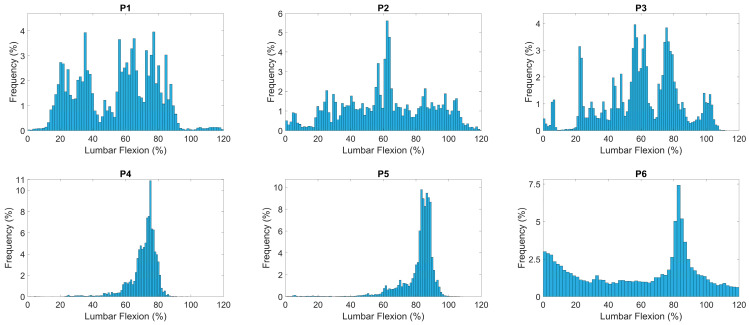
Amplitude probability distribution function (APDF) plots of the six participants through day postures.

**Figure 4 sensors-24-06751-f004:**
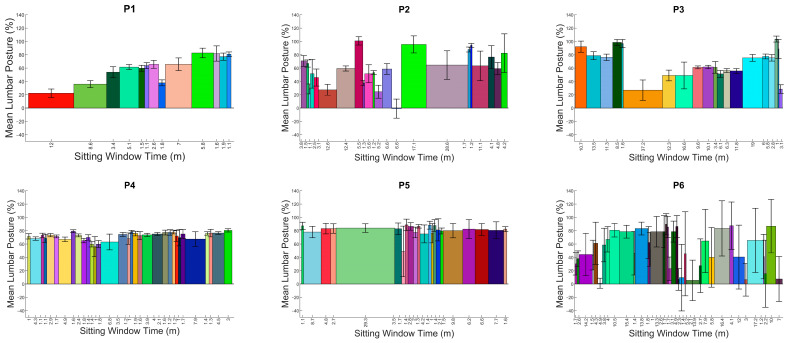
Through-day window-by-window analysis for each of the six participants in chronological order.

**Figure 5 sensors-24-06751-f005:**
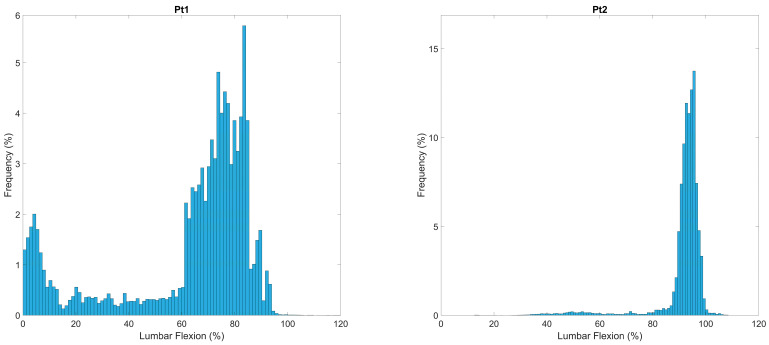
Amplitude probability distribution function (APDF) plots of the two participants diagnosed with LBP through-day postures.

**Figure 6 sensors-24-06751-f006:**
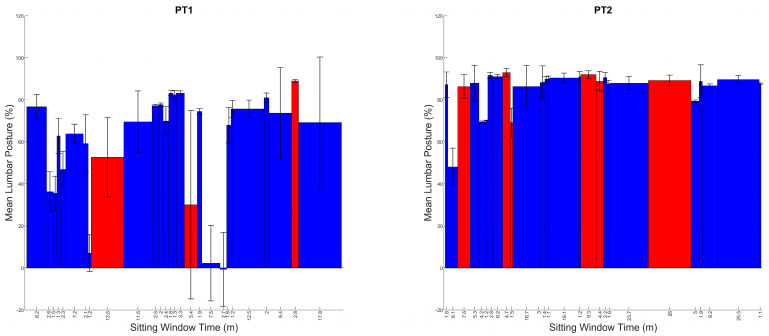
Through-day window-by-window analysis for each of the two participants with diagnosed LBP in chronological order. With non-painful windows in blue, and painful windows in red.

**Table 2 sensors-24-06751-t002:** Results taken from the accelerometer system to categorize each activity.

Sitting Region	Duration (min)	Mean Lumbar Flexion (°)	Mean Lumbar Flexion (%)	Standard Deviation (°)	Known Sitting Position	Sitting Identified by Algorithm
1	1.99	23.95	64.70	1.49	Normal sitting	Yes
2	1.99	40.04	108.20	1.59	Slouched sitting	Yes
3	2.01	33.57	90.71	3.99	Double cross-legged sitting slouched	Yes
4	2.01	21.75	58.78	1.20	Double cross-legged sitting erect	Yes
5	1.90	27.41	74.08	0.81	Single cross-legged sitting ankle to knee	No (Thigh angle < 60°)
6	1.94	28.79	77.79	1.93	Single cross-legged sitting normal	Yes
7	1.95	17.80	48.09	19.19	Slouched and erect sitting cycles	Yes
8	1.97	0.68	1.83	1.50	Erect sitting	Yes

°: degree; %: percentage of flexion range of motion.

**Table 3 sensors-24-06751-t003:** Day summary statistics for sitting for the six participants.

P	No. of Sitting Windows	Mean Sitting Window Duration (min) (SD)	Weighted Average Posture for the Day (%Flexion)	Average Posture for the Day (%Flexion)	Variability of Postures through Day (%Flexion)	Average Fidgeting through Day (%Flexion)	Weighted Average *SLPE*	Average *SLPE*	Variability of *SLPE* through Day
1	13	4.11 (3.27)	54.78 (23.60)	60.67	18.25	6.18	125.19	35.51	21.11
2	22	6.44 (6.47)	60.94 (26.54)	59.05	24.99	11.12	319.69	40.78	161.17
3	20	9.84 (7.90)	60.23 (23.91)	68.14	21.33	6.87	505.98	117.76	84.01
4	33	2.40 (1.68)	70.91 (8.93)	71.61	5.86	4.90	669.50	50.07	41.55
5	22	4.66 (5.97)	81.34 (11.67)	81.01	7.97	10.06	746.10	48.70	76.21
6	37	6.02 (5.03)	61.86 (37.34)	51.37	28.13	28.48	366.97	17.38	29.53

P: participant; No.: number; min: minutes; *SLPE*: sitting lumbar posture exposure; %Flexion: percentage of flexion range of motion.

**Table 4 sensors-24-06751-t004:** Baseline characteristics of the two patients for study 3.

Characteristic	Pt1	Pt2
Sex	M	F
Age (years)	53	35
Height (m)	1.71	1.81
Weight (kg)	85	86
Duration (years)	10	0.75
VAS (average)	54	18
VAS (worst)	63	42
FABQ	46	29
RMDQ	8	8
Aggravating Factors	Sitting Standing Walking	Sitting Standing

M: male; F: female; m: meters; kg: kilograms; VAS: visual analogue scale; FABQ: Fear-avoidance beliefs questionnaire; RMDQ: Roland–Morris disability questionnaire.

**Table 5 sensors-24-06751-t005:** Day summary statistics for the two participants with diagnosed LBP, including a breakdown of painful and non-painful windows.

P	Sitting Windows	No. of Sitting Windows	Mean Sitting Window Duration (min)	Weighted Average Posture for the Day (%Flexion)	Average Posture for the Day (%Flexion)	Variability of Postures through Day (%Flexion)	Average Fidgeting through Day (%Flexion)	Weighted Average *SLPE*	Average *SLPE*	Variability of *SLPE*
1	Overall	27	4.82	70.67	60.10	25.21	9.94	268.32	80.37	98.83
	Painful	3	7.26	61.86	57.28	24.28	21.46	70.82	135	162.21
	Pain Free	24	4.52	72.26	60.44	25.24	8.51	216.77	73.55	85.33
2	Overall	27	7.14	86.54	85.11	9.74	3.52	1745.8	324.37	330.64
	Painful	6	8.91	83.56	86.42	8.03	3.93	271.79	302.11	308.97
	Pain Free	21	6.63	87.23	84.73	10.16	3.41	1761.60	330.74	336.31

P: patient; No.: number; min: minutes; *SLPE*: sitting lumbar posture exposure; %Flexion: percentage of flexion range of motion.

## Data Availability

Data can be obtained by request from authors.

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
