# Peer review of "Lumbar Sitting Behavior of Individuals with Low Back Pain: A Preliminary Study Using Extended Real-World Data"

_sensors, 2024, doi:10.3390/s24206751_

Round 1
Reviewer 1 Report
Comments and Suggestions for Authors
Dear authors. I'm very impressed with your research. Registration of motor parameters in real life, something that doctors have long needed and is now becoming technically accessible. Yes, these are just the first steps, but in the right direction. The behavior patterns of the six subjects are as different as fingerprints. This is also a moment for subsequent analysis and the question of what are the boundaries of health. At the same time, the motor behavior of different healthy people can differ by orders of magnitude.
However, in your discussion you only touched on the main points related to your research. It seems reasonable to me to add to the discussion the preliminary characteristics of the first group examined. What do you have in common? Is it possible to talk about different motor types of behavior of men and women? Figure 3 gives the reader a lot to think about. And about this too. In the paragraph starting on line 139 or separately, I would recommend identifying and discussing the quantitative differences between the two examined patients and the six conditionally healthy ones. Their quantitative parameters are available only in the Abstract and Results.
In addition, after Figures 3, 4, 5 and 6 there are text repetitions. Not everywhere there is line numbering (it is also double), but for 5 and 6 these are lines 85-87.
Author Response
Reviewer 1 –
Dear authors. I'm very impressed with your research. Registration of motor parameters in real life, something that doctors have long needed and is now becoming technically accessible. Yes, these are just the first steps, but in the right direction. The behavior patterns of the six subjects are as different as fingerprints. This is also a moment for subsequent analysis and the question of what are the boundaries of health. At the same time, the motor behavior of different healthy people can differ by orders of magnitude.
Many thanks for your kind words regarding our study.
However, in your discussion you only touched on the main points related to your research. It seems reasonable to me to add to the discussion the preliminary characteristics of the first group examined. What do you have in common?
Many thanks for this comment, and the request to expand this section. We have now added a comparison between each of the participants varied behaviours, in the discussion.
Is it possible to talk about different motor types of behavior of men and women?
This would be great however; it is not possible to group the data into gender as each person seems to have an individualised characteristic evidenced in Figure 3/4/5/6. Perhaps something for our future work.
Figure 3 gives the reader a lot to think about. And about this too. In the paragraph starting on line 139 or separately, I would recommend identifying and discussing the quantitative differences between the two examined patients and the six conditionally healthy ones. Their quantitative parameters are available only in the Abstract and Results.
Many thanks for spotting this. The development and comparison have now been introduced into the discussion elaborating on the variability between ‘healthy’ and LBP participants.
In addition, after Figures 3, 4, 5 and 6 there are text repetitions. Not everywhere there is line numbering (it is also double), but for 5 and 6 these are lines 85-87.
Thanks for spotting this formatting error. We have done our best to correct it.
Reviewer 2 Report
Comments and Suggestions for Authors
1, the title is too long, please concise it.
2, Please remove 1-3 keyworks, the current is too much
3, How to get the data in Table 5?
4, How many sensors be used in Figure 2?
5, How to identify the activities? It seems that the activities different to various people, how to makt it clear?
6, Why choose IMU? Any reason for this brand? Any calibration?
7, is 6 participate enough for this research?
8, If the sensors in different location, does the reults the same? How about the different time and different status for the test?
9, Plenty of similar papers need to cote to solid the topic, such as Development trends and perspectives of future sensors and MEMS/NEMS. Micromachines. 11, 7, 541, 2020.
10, Does the results affect illness except low back pain? If different people, does the results suit for other people?
Comments on the Quality of English LanguageIt is fine for me
Author Response
Reviewer 2 –
1, the title is too long, please concise it.
Many thanks for highlighting this. We agree that the title could be condensed and have reduced it to:
Lumbar Sitting Behaviour of individuals with Low Back Pain: A Preliminary Study Using Through Day Real-World Data
2, Please remove 1-3 keyworks, the current is too much
We agree and have amended the key words. The new key words are:
Accelerometer, Spine, Posture, Variability, Pain, Acceptability
3, How to get the data in Table 5?
Sorry this was not immediately clear. Data collection procedure for the Table 5 data can be found on page 13, 6. Materials and Methods, starting on line 330.
4, How many sensors be used in Figure 2?
Sorry this seems to have been missed. Using the three sensor setup shown in figure 1, this has now been added into the figure 2 caption.
5, How to identify the activities? It seems that the activities different to various people, how to makt it clear?
Many thanks for this comment, however it is not immediately clear. We believe it relates to the clarity of people used for the different sections. For study 1 the activities are determined from each joint angle to verify the accuracy of the sensor readings, however in studies 2 and 3 the only determined activity is whether a participant is sitting or standing throughout their normal days activities, which was determined using a multi requirement clause in the code of the hip and thigh angle. We have re-structured the manuscript to add clarity.
6, Why choose IMU? Any reason for this brand? Any calibration?
Thanks for raising this. IMUs are the current state of the art for wireless tracking and the authorship have a strong track record in their use particularly pertaining to LBP. No calibration is needed, as we use raw data, and our method has been previously validated – as mentioned in this manuscript.
7, is 6 participate enough for this research?
We believe that 6 is enough for this preliminary study. We wanted to explore user experience and acceptability, and whether such a method has such an application to participants with LBP therefore this first study (of 9 people, 1 validation, 2 with LBP, 6 individuals) is needed.
8, If the sensors in different location, does the reults the same? How about the different time and different status for the test?
Many thanks for this great question. We do not know if the results would be the same as we have focussed on the proposed sensor locations for the lumbar spine. However, it is likely that the proposed analysis (histogram and ADPF etc.) would work with other sensor configurations.
9, Plenty of similar papers need to cote to solid the topic, such as Development trends and perspectives of future sensors and MEMS/NEMS. Micromachines. 11, 7, 541, 2020.
Many thanks for this suggestion. This has now been incorporated into the manuscript.
10, Does the results affect illness except low back pain? If different people, does the results suit for other people?
Many thanks again for this suggestion, similar to that above. There would be no reason that the approach couldn’t be used for determining sitting signatures of individuals with other conditions, for example, Stroke, Parkinson’s, Fibromyalgia etc. This is a measurement method for sitting behaviours which then would allow extrapolation or application to any condition. This has been added into the manuscript.
Reviewer 3 Report
Comments and Suggestions for Authors
This is a sound pilot study. The authors have taken the measurement into real world situation. This study includes only small number of people. Title should should reflect the pilot nature of the study.
Author Response
Reviewer 3 –
This is a sound pilot study. The authors have taken the measurement into real world situation. This study includes only small number of people. Title should reflect the pilot nature of the study.
Many thanks, we agree and the title has now been amended to fit the preliminary study design.
Reviewer 4 Report
Comments and Suggestions for Authors
The article is devoted to the study of the position detection system for back pain. The systems are based on the registration of the signal from the integrated accelerometer sensor.
Despite the relevant topic of the study, I suggest the authors to pay attention to the following points:
1. The abstract does not sufficiently correspond to the results of the work
2. The key words are chosen incorrectly
3. It is necessary to expand the analysis of the state of the problem, supplementing it with more relevant information
4. I suggest the authors to describe the structure of the registration system in the form of a block diagram
5. It is necessary to expand the information about the focus group
6. It is necessary to expand the information about the system limitations
7. What sampling frequency is used in the system?
8. Make a proofread of the text. There are a number of typos
9. Appendix A is redundant in this article
Author Response
Reviewer 4 –
- The abstract does not sufficiently correspond to the results of the work
Many thanks for drawing our attention to this. The abstract has been changed to reflect the themes of the manuscript more closely.
- The key words are chosen incorrectly
Many thanks for this comment. We have amended the keywords. They have been changed to Keywords: Accelerometer, Spine, Posture, Variability, Pain, Acceptability.
- It is necessary to expand the analysis of the state of the problem, supplementing it with more relevant information
The authors thank you for this comment. We feel that the introduction follows our preferred presentation of information. Paragraph 1 outlines LBP and its importance, paragraph 2, how sitting posture is related to LBP, paragraph 3 relates to sitting duration and the importance of monitoring this and overcoming the single point in time estimate, paragraph 4, introduces the limitations and opportunities of the technology and paragraph 5 the need for the elements of this study. We therefore feel this is the right balance of information to introduce our study.
- I suggest the authors to describe the structure of the registration system in the form of a block diagram
Many thanks for this comment. We feel that as this is a proprietary ‘off the shelf’ device, it is inappropriate to add the block diagram and it wouldn’t add anything to the purpose of the manuscript.
- It is necessary to expand the information about the focus group.
Many thanks for this comment. The details for the setup of the focus group being the six participants have been added, also extra information resulting from the discussion have been included in the results section for this too. This has really strengthened the manuscript.
- It is necessary to expand the information about the system limitations
This is a great comment. This has been added into the manuscript at the end of the discussion. The current method as proposed here draws on info from sagittal plane only. Therefore, information relative to other planes is not considered which could be relevant to other clinical conditions.
- What sampling frequency is used in the system?
The sampling frequency is 15Hz in the manuscript methods at line 118.
- Make a proofread of the text. There are a number of typos
Many thanks for identifying this. The manuscript has been proofread again to remove any typos or grammar issues.
- Appendix A is redundant in this article
This has now been removed from the manuscript.
Round 2
Reviewer 2 Report
Comments and Suggestions for Authors
No moren comments
Reviewer 4 Report
Comments and Suggestions for Authors
I received all the answers to my questions. All the necessary deficiencies were corrected